# ReFOCUS: Recurrent False Object Correction Using guidance Strategies in Object Detection

## Abstract

This work addresses the issue of recurrent false positive classification in object detection. We consider two experimental setups imitating real-world scenarios that lead to such errors: i) erroneous annotations, ii) non-objects that resemble actual objects. We show that resulting models can be corrected efficiently using a two-step protocol that leverages false positive annotations. For the first step, we present and compare two correction approaches that guide false positives toward true negatives, in either the latent or the logit space. The second step then consists in standard continuous fine-tuning on correct annotations. The latent guidance framework relies on a decoder that maps the bounding box of a given false positive to its target true negative embedding. The decoder is trained as part of an autoencoder, where appropriate true negative samples are generated by a learnable Gaussian mixture model in the latent space. By leveraging the properties of the Wasserstein distance, the mixture model is optimized through standard backpropagation. In both experimental setups, the two correction methods significantly outperform standard continuous fine-tuning on correct annotations and demonstrate competitive performance when compared to models retrained from scratch on correct annotations. In particular, in the second experimental setup, the latent guidance framework consistently outperforms these models, effectively enhancing detection performance at the cost of supplementary false positive annotations. Additionally, the proposed techniques prove effective in a few-shot learning context.

## 1 Introduction

The capacity to identify and locate objects is a fundamental aspect of computer vision, providing the basis for a vast array of applications, including autonomous driving, surveillance, robotics, and medical imaging. This process entails not only the identification of objects within an image but also precisely localizing them through bounding boxes. In recent years, significant advancements in deep learning, particularly with Vision Transformers (ViT) (Dosovitskiy et al., 2021), have greatly enhanced the accuracy and efficiency of object detection models. Despite these advancements, recurrent errors remain a major challenge, hindering the performance and generalization of these systems in industrial applications.

Errors in object detection can manifest in various forms, such as false positives (FPs), false negatives (FNs), misclassifications, and localization errors (Bolya et al., 2020). These errors often arise due to factors such as occlusion, varying object scales, complex backgrounds, or class imbalance in the training data. While localization errors stem from inaccurate bounding box predictions, other errors are typically the result of object misclassification. This study specifically addresses recurrent FP classification errors, where the model consistently detects an object that should not be identified, e.g. people on billboards as instances of real people. In this work, we examine two experimental setups that contribute to the occurrence of such recurrent errors:

    i) A model $f_{\text{Noisy}}$ trained on a noisy dataset $\mathcal{D}_{\text{Noisy}}$, where certain instances of recurrent FPs are incorrectly labeled as objects.

ii) A model $f_{\text{True}}$ trained on a well-annotated dataset $\mathcal{D}_{\text{True}}$, where instances of recurrent FPs are rightfully not annotated. However, these instances bear resemblance to another object that is to be detected, causing $f_{\text{True}}$ to misidentify them as true objects.

The objective of this work is not only to correct the FP errors caused by one of the aforementioned conditions, but simultaneously to ensure that these corrections do not negatively impact the model's performance on the rest of the dataset. By addressing these recurrent errors, our aim is to enhance the overall performance of the model. Ultimately, this study seeks to contribute to the field of object detection by introducing methodologies and insights that can be generalized across different datasets and detection frameworks.

Our research is based on DETR model (Carion et al., 2020). For the correction process, we assume that we have access to a corrective dataset $\mathcal{D}_{\text{C}}$, which is a subset of the correct dataset $\mathcal{D}_{\text{True}}$ where recurrent FPs are additionally annotated as 'FP'.

**Motivations**    To improve $f_{\text{Noisy}}$, one solution would be to retrain the model from scratch on $\mathcal{D}_{\text{True}}$. Nevertheless, this approach is often impractical due to excessive computational time. Furthermore, there are cases where the whole original training data may no longer be available. Therefore, a more viable alternative is to develop correction frameworks based on continuous fine-tuning of the learned model, preserving the knowledge gained from previous training data. Moreover, in the case of $f_{\text{True}}$, retraining on $\mathcal{D}_{\text{True}}$ would be ineffective as it was already trained on this dataset in the first place.

**Contributions**    We propose two innovative correction frameworks that guide FPs toward TNs in either the latent space or the logit space. The latent guidance framework leverages an autoencoder where a learnable Gaussian mixture model generates the embeddings of appropriate TNs, and a straightforward decoder retrieves the TN embedding given a bounding box. We utilize the properties of the Wasserstein distance to train the Gaussian mixture model through standard backpropagation. Finally, we assess and compare the outcomes across these two distinct spaces.

## 2    RELATED WORK

Our work builds upon and draws inspiration from a range of research areas, including object detection, contrastive learning, and machine unlearning.

**Object Detection**    Object detection is a well-explored field in Computer Vision. Traditional detectors, such as HOG (Dalal & Triggs, 2005) and DPM (Felzenszwalb et al., 2010), relied on handcrafted image features as priors. However, the advent of end-to-end neural network-based methods revolutionized the field over the past decade, beginning with Convolutional Neural Networks (CNN) (Krizhevsky et al., 2012), and more recently, Vision Transformers (ViT) (Dosovitskiy et al., 2021). Unlike CNN, which rely on convolutional layers, ViT use attention mechanisms to capture global dependencies across an image. Modern deep learning-based object detectors can be broadly categorized into two main architectures: single-stage detectors, such as YOLO (Redmon et al., 2016) and DETR (Carion et al., 2020), and two-stage detectors, like those based on the R-CNN family of models (Girshick et al., 2014).

**DETR**    In 2020, Carion et al. (2020) introduced an innovative one-stage architecture that leverages ViT. After a ResNet backbone (He et al., 2015), the image features are extracted and then processed by a transformer encoder, which captures global dependencies across the entire image through self-attention. They are then used for cross-attention in the decoder. The decoder takes $N$ learnable object queries as input and applies a series of self-attention and cross-attention mechanisms with the encoder's image feature embeddings. The result is $N$ potential objects in a latent space $\mathcal{Z}$. Two Multi-Layer Perceptrons (MLP), $\text{MLP}_{\text{class}}$ and $\text{MLP}_{\text{bbox}}$, then map each potential object to a class label and a bounding box, as illustrated in Fig. 1. Since the number of object queries $N$ is fixed, the model can predict a 'no-object' class, denoted as $\varnothing$.

After a bipartite matching, ensuring a one-to-one correspondence between predicted and ground truth objects (or $\varnothing$ for no-object predictions), the loss of DETR consists of two components: one for classification and another for localization accuracy. Following the notations from the original paper:

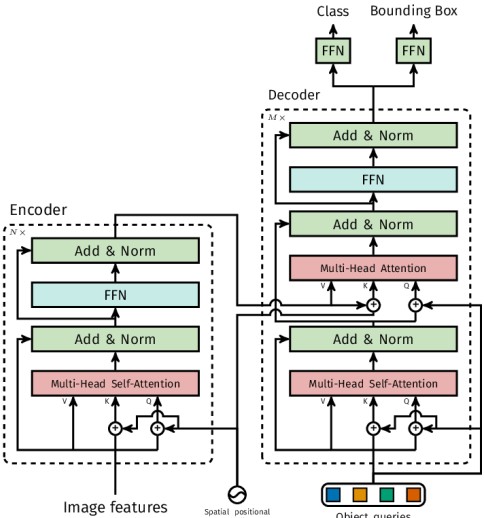

Figure 1: Architecture of DETR's transformer. Taken from the original paper (Carion et al., 2020).

$$\mathcal{L}_{\text{DETR}}(y, \hat{y}) = \sum_{i=1}^{N} \left[ -\log \hat{p}_{\hat{\sigma}(i)}(c_i) + \mathbf{1}_{\{c_i \neq \varnothing\}} \left( \lambda_{\text{iou}} \mathcal{L}_{\text{iou}}(b_i, \hat{b}_{\hat{\sigma}(i)}) + \lambda_{\text{L1}} ||b_i - \hat{b}_{\hat{\sigma}(i)}||_1 \right) \right] \quad (1)$$

**Machine Unlearning**    Machine Unlearning (MU) serves as a data forgetting mechanism that aligns with regulations like *"The Right to be Forgotten"* under GDPR (Zhang et al., 2024). It aims to adjust a trained model so that it behaves as though certain data has never been encountered, thereby preserving performance while facilitating the removal of specific samples. Applications of this concept, particularly in class forgetting, are explored by Tarun et al. (2024), who propose a framework utilizing data augmentation. This method involves an *impair step* to unlearn the forget classes, followed by a *repair step* to restore accuracy on retained classes. There is a key distinction between MU and our approach: while MU aims to forget specific samples, our objective is to generalize across all samples of the same type.

**Contrastive Learning**    The objective of contrastive learning is to construct an embedding space where similar samples are close together and dissimilar samples are farther apart. A wealth of research has developed various frameworks and methodologies with different loss functions, yielding increasingly sophisticated results (Schroff et al., 2015; Sohn, 2016; Chen et al., 2020). Initially effective in unsupervised and self-supervised contexts, contrastive learning has also shown success in supervised learning (Radford et al., 2021; Khosla et al., 2021). Most recent studies utilize a similar loss function, analogous to a cross-entropy loss in the embedding space.

**False Positive Suppression**    Cheng et al. (2020) propose decoupling classification refinement from localization tasks, utilizing one model for bounding box predictions and another for class predictions based on these candidates. This approach transforms the model into a two-stage detector, which is not our objective here. Chen et al. (2020) tackle the problem of FPs in domain adaptation for pedestrian detection. Their key contribution is the introduction of an unsupervised re-ranking mechanism that clusters bounding boxes and re-ranks them to suppress FPs, addressing domain shifts without the need for annotated data. Although impressive, their results still lag behind oracle models trained on annotated data.

## 3 METHOD

### 3.1 CORE CONCEPT

Two correction frameworks are developed, both based on the same underlying concept of shifting FPs toward TNs. From an optimization perspective, this can be achieved by minimizing the discrepancy between FPs and TNs. We focus on two distinct spaces for that. The first is the logit space, which directly influences the model's class predictions. The second is the latent space $\mathcal{Z}$, situated one level deeper, prior to the classifier $\text{MLP}_{\text{class}}$ and the bounding box predictor $\text{MLP}_{\text{bbox}}$. This space is deemed pertinent because it encompasses all the extracted features necessary for both class prediction and localization prediction of a potential object.

Our approach involves two steps and draws inspiration from the methodology introduced by Tarun et al. (2024). The process begins with a guide step, where, notably, FPs moved toward TNs by minimizing Eq. 2. This step updates the model's weights, which may inadvertently affect the accuracy for the remaining objects. To mitigate this, we introduce a repair step to restore the performance on the remaining objects, achieved by fine-tuning the model on $\mathcal{D}_{\text{True}}$ using only the standard DETR loss.

$$\mathcal{L}_{\text{Guide}}(\boldsymbol{y}, \hat{\boldsymbol{y}}) = \lambda_{\text{Correct}} \sum_{\hat{y_i} \in \text{FP}} \mathcal{L}_{\text{Correct}}(\hat{y_i}) + \lambda_{\text{DETR}} \sum_{\hat{y_j} \notin \text{FP}} \mathcal{L}_{\text{DETR}}(y_j, \hat{y_j}) \tag{2}$$

### 3.2 LoGF: LOGIT GUIDANCE FRAMEWORK

We begin by developing the Logit Guidance Framework (LoGF), which aims to transform FPs into TNs within the logit space. To convert FPs into TNs, the cross-entropy (CE) loss is utilized:

$$\mathcal{L}_{\text{Correct}}(\hat{y}) = \mathcal{L}_{\text{CE}}(\varnothing, \hat{y}) \tag{3}$$

It is noteworthy that this term is already included in $\mathcal{L}_{\text{DETR}}$. However, Carion et al. (2020) down-weight the log-probability term associated with the 'no-object' class to mitigate class imbalance, as only a small fraction of the numerous potential objects are actual objects. Consequently, this correction specifically up-weights the cross-entropy associated with FPs in $\mathcal{D}_{\text{C}}$, thereby increasing their significance.

### 3.3 LaGF: LATENT GUIDANCE FRAMEWORK

#### 3.3.1 AUTOENCODER

We continue by developing the Latent Guidance Framework (LaGF), which aims to guide FPs toward TNs within the latent space $\mathcal{Z}$. The objective is to establish a more effective clustering structure that assists the classifier in making more accurate predictions.

Unlike the logit space, where the target TN to which all the FPs should move is straightforward ($\varnothing$), $\mathcal{Z}$ is a high-dimensional space where many points represent TNs.

Since $\mathcal{Z}$ encodes both the class and the bounding box of an object simultaneously, objects that share similarities (class and bounding box) should be proximate in this space. Therefore, a suitable TN candidate for a given FP is one that shares the greatest similarity with that FP. Given that the class of the TN is already determined, only its position remains to be defined. We can define an appropriate TN for a FP as follows:

**Definition 1** *[Target $\epsilon$-TN for a FP] Let $z_{FP} \in \mathcal{Z}$ represent the embedding of a FP and $y_{bbox}$ denote its bounding box prediction:*
$$y_{bbox} = \sigma \circ MLP_{bbox}(z_{FP})$$

*Let $z_{TN} \in \mathcal{Z}$. $z_{TN}$ is the embedding of a target $\epsilon$-TN for $z_{FP}$ if $z_{TN}$ satisfies the following conditions:*
$$[Softmax \circ MLP_{class}(z_{TN})]_{\varnothing} \geq 1 - \epsilon \quad and \quad \sigma \circ MLP_{bbox}(z_{TN}) = y_{bbox} \tag{4}$$

In other words, a target $\epsilon$-TN for a given FP is a sample that belongs to the 'no-object' class $\varnothing$ with a probability of at least $1 - \epsilon$ and shares the same bounding box as the FP. In the following, we will omit $\epsilon$, implying that we are seeking target TNs with the highest probability.

An autoencoder architecture is proposed to identify the embeddings of target TNs for all FPs in $\mathcal{D}_C$, which are close to those of the FPs. This autoencoder is based on a Gaussian Mixture Model (GMM) that learns to generate embeddings of target TNs in $\mathcal{Z}$ for similar FPs in $\mathcal{D}_C$. Following the GMM, a decoder learns to retrieve the embedding in $\mathcal{Z}$ of a target TN given a bounding box. The end-to-end architecture described in Fig. 2a addresses the variance issues detailed in A.3.

Once the decoder $\Phi$ is trained, the embedding $z^f \in \mathcal{Z}$ of a given FP $\hat{y}$ is guided toward the embedding of a close target TN $\tilde{z} = \Phi \circ \sigma \circ \mathrm{MLP}_{\mathrm{bbox}}(z^f)$ using the following loss:

$$\mathcal{L}_{\mathrm{Correct}}(\hat{y}) = \log\left(1 + \exp\left(-\frac{sim(z^f, \tilde{z})}{T}\right)\right) \quad \text{with} \quad sim(z^f, \tilde{z}) = \frac{\langle z^f, \tilde{z} \rangle}{\|z^f\|\|\tilde{z}\|} \qquad (5)$$

where $T$ is a hyperparameter and $sim$ denotes the cosine similarity. The schema for the guide step is illustrated in Fig. 2b. Note that the distinction between FP embeddings $\{z_i^f\}_i$ and remaining embeddings $\{z_j^r\}_j$ is obtained using a bipartite matching after the forward pass.

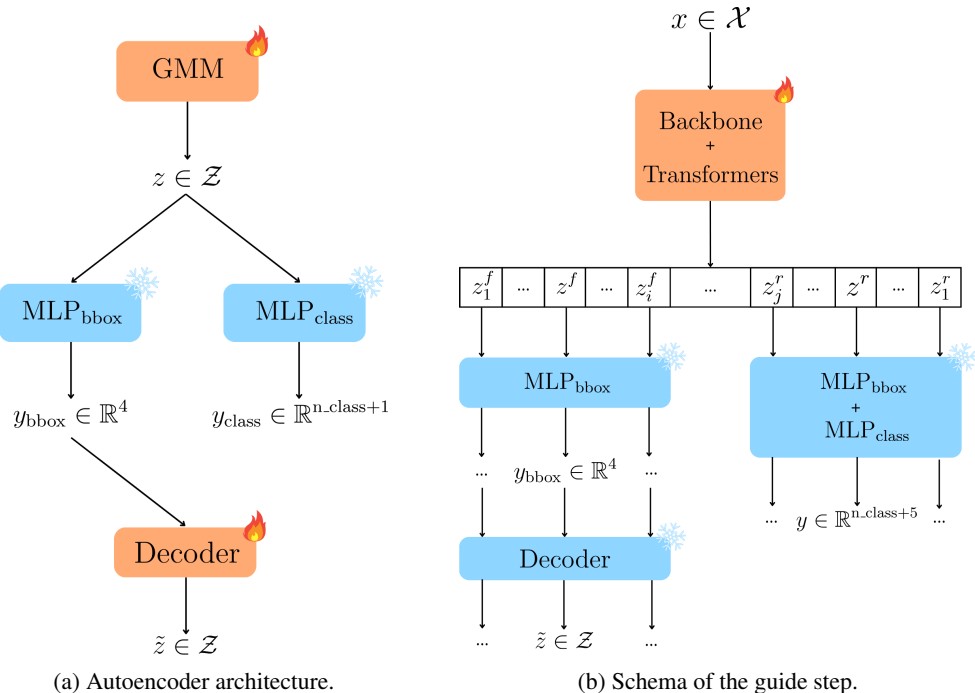

(a) Autoencoder architecture.     (b) Schema of the guide step.

Figure 2: Architectures used for the latent guiding framework.

Note: During the guide step, FPs are converted into TNs, though not always with high confidence in early updates. LaGF aims to progressively guide these samples toward TNs with increasing confidence, until they reach a fixed point of the decoder. In this way, LaGF establishes an elegant and natural displacement that facilitates correction.

### 3.3.2 GAUSSIAN MIXTURE MODEL

The purpose of the Gaussian Mixture Model (GMM) is to generate samples in $\mathcal{Z}$ that are embeddings of target TNs for similar FPs in $\mathcal{D}_C$, which are close to the embeddings of FPs in $\mathcal{D}_C$, in order to train the decoder effectively. We employ a GMM due to its property as a universal approximator of smooth densities (Goodfellow et al., 2016).

Firstly, the GMM must generate TNs samples, which is ensured by:

$$\mathcal{L}_{\text{TN}}(z) = \mathcal{L}_{\text{CE}}\left(\varnothing, \text{MLP}_{\text{class}}(z)\right) \tag{6}$$

Secondly, the parameters of the GMM should be trained such that the image by $\sigma \circ \text{MLP}_{bbox}$ of its distribution in $\mathcal{Z}$ is similar to the distribution of the bounding boxes of FPs in $\mathcal{D}_{\text{C}}$. To measure the similarity between these two distributions, the Wasserstein distance in the discrete case is employed, since it is a very natural and canonical distance to optimize (Peyré & Cuturi, 2020). The Wasserstein distance between two discrete distributions $\alpha = \sum_{i=1}^{n} a_i \delta_{x_i}$ and $\beta = \sum_{j=1}^{m} b_j \delta_{y_j}$, considering the Euclidian distance as metric, is defined as follows:

$$\mathcal{W}(\alpha, \beta) = \min_{\Pi \in U(\alpha, \beta)} \sum_{i=1}^{n} \sum_{j=1}^{m} \|x_i - y_j\| \Pi_{i,j} \tag{7}$$

where $U(\alpha, \beta)$ is the set of joint probability distributions with marginals $\alpha$ and $\beta$ such that $U(\alpha, \beta) = \{\Pi \in \mathbb{R}_+^{n \times m} \mid \Pi \mathbf{1}_m = \boldsymbol{a} \text{ and } \Pi^\top \mathbf{1}_n = \boldsymbol{b}\}$ and $\Pi$ is the transport plan that enables the movement from $\alpha$ to $\beta$.

Consider $\alpha$ as the discrete distribution defined as the image by $\sigma \circ \text{MLP}_{\text{bbox}}$ of the samples generated by the GMM, and $\beta$ as the target discrete distribution of bounding boxes of FPs in $\mathcal{D}_{\text{C}}$. In order to transport $\alpha$ toward $\beta$ while avoiding the centres of mass, it is necessary to sample the same number of points in $\alpha$ as in $\beta$ (Peyré & Cuturi, 2020), as done by Arjovsky et al. (2017).

Let $Z \sim \text{GMM}\left(\{\pi_i, \mu_i, \Sigma_i\}_{i=1}^{K}\right)$ be a random variable in $\mathcal{Z}$ that follows the distribution of a mixture of $K$ Gaussians. Let $N$ be the number of samples of our target distribution and $m$ the number of points of our generated distribution. Now let $U \sim \mathcal{U}(\{1, .., N\})$ be a random variable that follows a uniform distribution in $\{1, .., N\}$. Let $z_1, ..., z_m$ and $u_1, ..., u_m$ be $m$ samples from $Z$ and $U$ respectively. The loss to optimise is the following:

$$\mathcal{L}_{\text{bbox-dist}} = \mathcal{W}\left(\frac{1}{m} \sum_{i=1}^{m} \delta_{\sigma \circ \text{MLP}_{\text{bbox}}(z_i)}, \frac{1}{m} \sum_{i=1}^{m} \delta_{y_{u_i}^{\text{FP}}}\right) \tag{8}$$

where $\{y_i^{\text{FP}}\}_{i=1}^{N}$ is the set of bounding boxes of FPs in $\mathcal{D}_{\text{C}}$.

Thirdly, to ensure training stability by selecting target TN embeddings close to FP ones, we initialize the GMM according to the initial latent distribution of FPs in $\mathcal{D}_{\text{C}}$ using the Expectation-Maximization (EM) algorithm (Dempster et al., 2018).

In summary, the loss that contributes to training the GMM within the total autoencoder loss is:

$$\mathcal{L}_{\text{GMM}} = \mathcal{L}_{\text{bbox-dist}} + \mathcal{L}_{\text{TN}} \tag{9}$$

**Weight reparameterization trick** In order to minimise the aforementioned loss with respect to the parameters of the GMM ($\{\pi_i, \mu_i, \Sigma_i\}_{i=1}^{K}$), gradient descent and classic backpropagation are employed. Nevertheless, it is not possible to perform backpropagation through a sample derived from a random variable $\mathbf{X}$. Kingma & Welling (2022) proposed a reparameterization trick to sample from a normal distribution. However, to sample from a GMM, it is first necessary to sample from a categorical distribution to select which Gaussian should be sampled. This is a known issue that Graves (2016) tackles for all continuous multivariate distributions with a differentiable density function by computing explicitly the backpropagation over the weights. Nevertheless, we propose an alternative method using conventional backpropagation that leverages the properties of Wasserstein distance, which allows weighting of all data points within the histogram. Hence, we sample the same number of points per Gaussian and then apply weighting within the histogram of the corresponding bounding box distribution. More precisely, if we sample $m$ points from the $i$-th Gaussian, each sample will be weighted by a factor of $\frac{\pi_i}{m}$ within the histogram. The weight reparameterization trick is detailed in the autoencoder training algorithm described in A.1 and the proof is provided in A.2.

### 3.3.3 DECODER

The decoder is a function $\Phi : \mathbb{R}^4 \to \mathcal{Z}$ that takes the bounding box of a FP as input and outputs the embedding of a target TN for that FP. A straightforward neural network is employed as the decoder; further details regarding the network architecture can be found in A.4. The training of the decoder is conducted by minimizing the following loss function:

$$\mathcal{L}_{\text{decoder}}\left(y_{\text{bbox}}, z\right) = \mathcal{L}_{\text{MSE}}\left(\Phi(y_{\text{bbox}}), z\right) \tag{10}$$

where $z \in \mathcal{Z}$ is a TN embedding generated by the GMM and $y_{\text{bbox}} = \sigma \circ \text{MLP}_{\text{bbox}}(z)$. Since the decoder's parameters are optimized with fixed MLPs, the utilization of the decoder during the guide step necessitates freezing $\text{MLP}_{\text{class}}$ and $\text{MLP}_{\text{bbox}}$.

## 4 EXPERIMENTAL SETUP

### 4.1 DATASET

For the experiments, we use the PASCAL VOC 2007 dataset that comprises 20 object classes, including animals, vehicles, person and indoor categories of objects (Everingham et al.). In the next sections, all datasets have the same support, only the annotations change. In a first dataset, we intentionally misannotate all buses as cars, and in a second dataset, we misannotate all sofas as chairs, given the notable similarities between these pairs of objects. This misannotation of buses (resp. sofas) leads to the creation of the noisy dataset $\mathcal{D}_{\text{Noisy}}$, on which we train $f_{\text{Noisy}}$. In contrast, we train $f_{\text{True}}$ on the well-labeled dataset $\mathcal{D}_{\text{True}}$ where buses (resp. sofas) are not annotated at all.

Additionally, two experimental cases are investigated: a single-class case that only considers the confused-class objects (cars or chairs) and a multi-class case in which all classes are present. The single-class case allows for a more detailed examination of the impact of the correction while maintaining good performance on the confused class. The multi-class case also enables the assessment of performance changes in other classes that are not directly affected by the correction.

### 4.2 METHODS TO COMPARE

We summarize all classes of experiments in Tab. 1. Some of the aforementioned methods allow continuous fine-tuning whereas the others entail retraining from scratch. In particular, we train an oracle model $f_{\text{Oracle}}$ from scratch which identifies and classifies the FPs into a new dedicated class. $f_{\text{True}}$ and $f_{\text{Oracle}}$ serve as target models, while we benchmark our methods to classic continuous fine-tuning.

Table 1: Models and methods of correction.

| Corrected model | Initial model | Methods of correction | Continous fine-tuning |
|---|---|---|---|
| $f_{\text{Noisy}}^{\text{Fine-tune}}$ | $f_{\text{Noisy}}$ | Fine-tuning on the correct dataset $\mathcal{D}_{\text{True}}$ | ✓ |
| $f_{\text{Noisy}}^{\text{LoGF}}$ (Ours) | $f_{\text{Noisy}}$ | Guiding in the logit space (LoGF) | ✓ |
| $f_{\text{Noisy}}^{\text{LaGF}}$ (Ours) | $f_{\text{Noisy}}$ | Guiding in the latent space $\mathcal{Z}$ (LaGF) | ✓ |
| $f_{\text{True}}^{\text{LoGF}}$ (Ours) | $f_{\text{True}}$ | Guiding in the logit space (LoGF) | ✓ |
| $f_{\text{True}}^{\text{LaGF}}$ (Ours) | $f_{\text{True}}$ | Guiding in the latent space $\mathcal{Z}$ (LaGF) | ✓ |
| $f_{\text{True}}$ | - | Training from scratch on the correct dataset $\mathcal{D}_{\text{True}}$ | ✗ |
| $f_{\text{Oracle}}$ | - | Training from scratch with FP as a new target class | ✗ |

### 4.3 METRICS

To assess the performance of the model before and after the correction, we use mean Average Precision (mAP). mAP averages precision over different recall thresholds for each class, and then averages it across all classes. mAP is also averaged at various IoU thresholds (from 0.5 to 0.95

by steps of 0.05) to take into account the localization accuracy. In the single-class case, we also compute precision and recall to assess the correction of FPs and the deterioration over TPs. We choose the confidence threshold that maximizes the F1-score over the validation set. Precision and recall are computed at a low IoU threshold (0.5) to avoid biases in localization and FN errors (Bolya et al., 2020). In the multi-class case, we compute the Average Precision (AP) on the confused class to assess the effectiveness of the correction of the FPs, and mAP on the other classes to evaluate the incidental impact of our methods on these classes.

## 5  EXPERIMENTAL RESULTS

### 5.1  RESULTS

We use a DETR model with a ResNet-50 backbone, trained end-to-end on MS-COCO 2017 (Lin et al., 2015) as a base model. We present the results in the single-class case in Tab. 2 and Tab. 3, and the results in the multi-class case in Tab. 4. In all cases, five runs were conducted for each model, except for the model to correct.

**i) Correction of $f_{\text{Noisy}}$**  In both single-class and multi-class cases, our correction frameworks demonstrate superior performance in mAP compared to the standard continuous fine-tuning method, highlighting the importance of FP guidance in the first step. The classic continuous fine-tuning method can, in fact, be viewed as a version of our correction framework without the guide step. The high improvement in precision in the single-class case with our correction frameworks ($\sim +13$ and $\sim +18$) compared to the improvement with the classic continuous fine-tuning method ($\sim +2$ and $\sim +5$) proves the efficiency of the suppression of FPs using the guidance framework. Therefore, the guide step can be seen as a powerful correction step. The evolution of recall across the different methods indicates that LoGF excessively increases FNs, unlike the continuous fine-tuning method and LaGF. In addition to demonstrating the superiority of guidance methods over continuous fine-tuning in terms of error correction, the multi-class case shows that none of the methods (including guidance-based approaches) interfere with the detection of objects from other classes. Furthermore, LaGF consistently outperforming the target models $f_{\text{True}}$ and $f_{\text{Oracle}}$ in terms of mAP, proving more stable than the other correction methods.

**ii) Correction of $f_{\text{True}}$**  Likewise, LaGF consistently improves the initial model $f_{\text{True}}$, enhancing its overall performance. In contrast, LoGF reduces the performance of $f_{\text{True}}$ in Tab. 2, indicating that LoGF is less reliable than LaGF. An inspection of the precision and recall values indicates that the overall improvements stem from the correction of annotated FPs.

Table 2: Results in the single-class case: bus and car.

| | mAP | | Precision | | Recall | |
|---|---|---|---|---|---|---|
| | mean | std | mean | std | mean | std |
| $f_{\text{True}}$ | 64.46 | 1.07 | 87.65 | 1.89 | 84.14 | 1.83 |
| $f_{\text{Oracle}}$ | 63.88 | 1.30 | 86.39 | 2.82 | 85.93 | 2.70 |
| $f_{\text{Noisy}}$ [1] | 61.39 | – | 75.62 | – | 85.14 | – |
| $f_{\text{Noisy}}^{\text{LoGF}}$ | **65.18** ↑ | **0.22** | **88.46** ↑ | **1.99** | 82.18 ↓ | 2.28 |
| $f_{\text{Noisy}}^{\text{LaGF}}$ | 64.68 ↑ | 0.38 | 88.17 ↑ | 1.15 | 83.31 ↓ | 1.16 |
| $f_{\text{Noisy}}^{\text{Fine-tune}}$ | 63.31 ↑ | 0.61 | 77.70 ↑ | 0.63 | **84.94** ↓ | **0.95** |
| $f_{\text{True}}$ [1,2] | 65.08 | – | 85.73 | – | 86.18 | – |
| $f_{\text{True}}^{\text{LoGF}}$ | 64.86 ↓ | 0.48 | 86.29 ↑ | 0.56 | 85.04 ↓ | 0.75 |
| $f_{\text{True}}^{\text{LaGF}}$ | **65.53** ↑ | **0.32** | **86.50** ↑ | **0.48** | 85.71 ↓ | 0.75 |

### 5.2  FEW-SHOT

To clearly observe the evolution of performance with respect to the size of the training set, we conduct few-shot experiments in the single-class case using the bus-car dataset, starting from $f_{\text{noisy}}$.

---

[1]This serves as the initial model to correct.

[2]Chosen randomly among the 5 runs.

Table 3: Results in the single-class case: sofa and chair.

| | mAP | | Precision | | Recall | |
|---|---|---|---|---|---|---|
| | mean | std | mean | std | mean | std |
| $f_{\text{True}}$ | 37.38 | 1.17 | 72.15 | 4.07 | 62.84 | 3.68 |
| $f_{\text{Oracle}}$ | 35.51 | 1.64 | 70.22 | 3.57 | 59.67 | 3.97 |
| $f_{\text{Noisy}}^{1}$ | 27.89 | – | 56.77 | – | 59.53 | – |
| $f_{\text{Noisy}}^{\text{LoGF}}$ | 37.10 ↑ | 1.50 | 74.16 ↑ | 0.96 | 55.44 ↓ | 2.54 |
| $f_{\text{Noisy}}^{\text{LaGF}}$ | **38.62** ↑ | **0.33** | **75.16** ↑ | **1.93** | **60.09** ↑ | **1.93** |
| $f_{\text{Noisy}}^{\text{Fine-tune}}$ | 31.90 ↑ | 2.89 | 61.63 ↑ | 4.61 | 56.90 ↓ | 4.75 |
| $f_{\text{True}}^{12}$ | 39.22 | – | 72.38 | – | 64.26 | – |
| $f_{\text{True}}^{\text{LoGF}}$ | **39.64** ↑ | **0.20** | 75.73 ↑ | 1.99 | **60.74** ↓ | **1.72** |
| $f_{\text{True}}^{\text{LaGF}}$ | 39.31 ↑ | 0.54 | **75.79** ↑ | **2.37** | 60.48 ↓ | 1.90 |

Table 4: Results in the multi-class case: bus and car (left), sofa and chair (right).

| | AP confused class | | mAP other classes | |
|---|---|---|---|---|
| | mean | std | mean | std |
| $f_{\text{True}}$ | 62.40 | 1.17 | 54.17 | 0.43 |
| $f_{\text{Oracle}}$ | 63.26 | 0.76 | 54.60 | 0.45 |
| $f_{\text{Noisy}}^{1}$ | 57.45 | – | 55.10 | – |
| $f_{\text{Noisy}}^{\text{LoGF}}$ | 61.34 ↑ | 0.71 | **55.12** ↑ | **0.18** |
| $f_{\text{Noisy}}^{\text{LaGF}}$ | **63.30** ↑ | **0.52** | 54.88 ↓ | 0.17 |
| $f_{\text{Noisy}}^{\text{Fine-tune}}$ | 59.36 ↑ | 1.63 | 54.46 ↓ | 0.22 |
| $f_{\text{True}}^{12}$ | 61.38 | – | 53.45 | – |
| $f_{\text{True}}^{\text{LoGF}}$ | 63.14 ↑ | 0.22 | 55.06 ↑ | 0.19 |
| $f_{\text{True}}^{\text{LaGF}}$ | **63.37** ↑ | **0.22** | 55.11 ↑ | 0.30 |

| | AP confused class | | mAP other classes | |
|---|---|---|---|---|
| | mean | std | mean | std |
| $f_{\text{True}}$ | 35.97 | 0.60 | 55.60 | 0.44 |
| $f_{\text{Oracle}}$ | 35.87 | 1.57 | 55.02 | 0.79 |
| $f_{\text{Noisy}}^{1}$ | 29.28 | – | 55.53 | – |
| $f_{\text{Noisy}}^{\text{LoGF}}$ | 35.38 ↑ | 0.78 | 55.26 ↓ | 0.41 |
| $f_{\text{Noisy}}^{\text{LaGF}}$ | **37.15** ↑ | **0.42** | **55.84** ↑ | **0.32** |
| $f_{\text{Noisy}}^{\text{Fine-tune}}$ | 33.78 ↑ | 1.68 | 55.03 ↓ | 0.26 |
| $f_{\text{True}}^{12}$ | 35.36 | – | 56.01 | – |
| $f_{\text{True}}^{\text{LoGF}}$ | 35.51 ↑ | 0.29 | 56.55 ↑ | 0.16 |
| $f_{\text{True}}^{\text{LaGF}}$ | **35.63** ↑ | **0.26** | **56.57** ↑ | **0.21** |

The results are presented in Fig. 3. We do not observe a significant decrease in the performance of any particular model: They all appear stable, with a slight preference for LaGF and the continuous fine-tuning method. Even with 10% of the training set, LaGF improves the mAP of the initial model by about 2 points, reaching around 60% of the maximum improvement seen when using the entire training set.

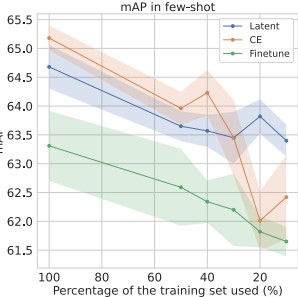 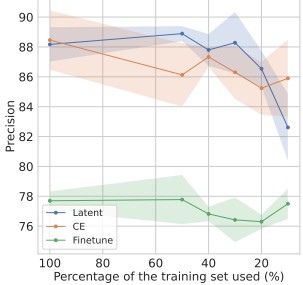 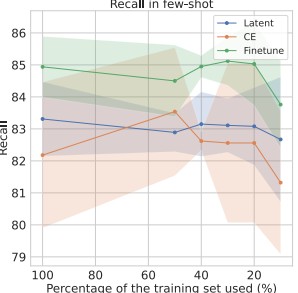

Figure 3: Results in a few-shot context.

# 6 ABLATION STUDIES

We conduct ablation studies to emphasize the significance of main design choices within LaGF, which has proven to be the most consistent and effective correction approach. These studies are carried out in a multi-class setting using the bus-car dataset, starting from $f_{\text{noisy}}$. The results are presented in Tab. 5.

First, we analyze the impact of removing the repair step. Results show that the detection performance is degraded across all classes, including the confused class. While the guide step improves the

clustering of the confused class in $\mathcal{Z}$, unfreezing the MLPs during the repair step appears essential for effective correction.

Next, we examine the effect of removing the DETR loss during the guide step. Interestingly, this has no significant impact on detecting objects from other classes, but it leads to a sharp decline in AP for the confused class. This can be attributed to the fact that using the DETR loss helps preserve the model's performance across other classes during the guide step, preventing drastic changes during the repair step that could otherwise undermine the corrections made during the guide step.

Table 5: Ablation studies using the latent guidance framework in the multi-class case: bus and car.

| # | Description | AP confused class | | mAP other classes | |
|---|---|---|---|---|---|
| | | mean | std | mean | std |
| – | Initial model to correct $f_{\text{Noisy}}$ | 57.45 | – | 55.10 | – |
| – | Reference model $f_{\text{Noisy}}^{\text{LaGF}}$ | 63.30 | 0.52 | 54.88 | 0.17 |
| 1 | Remove the repair step | 61.35 | 0.70 | 52.65 | 0.26 |
| 2 | $\mathcal{L}_{\text{Guide}} = \mathcal{L}_{\text{Correct}}$ | 61.11 | 0.40 | 55.03 | 0.24 |

## 7 CONCLUSION

We introduced two novel frameworks for correcting recurrent FP classification errors in object detection. These frameworks focus on guiding FPs toward TNs in either the latent space or the logit space. Identifying suitable TNs in the latent space proved to be a challenging task, which we addressed by using an autoencoder architecture that incorporates a learnable GMM and a straightforward decoder. By applying these correction mechanisms, we achieved significant and consistent improvements in model performance, with the latent guidance framework being especially effective.

Future work could extend the latent guidance framework to other object detection architectures. Additionally, an interesting direction would be to correct other recurring confusion errors, where instances of one class are often misclassified as another. In such cases, the framework could guide these FPs back to their true class, rather than to $\varnothing$. Moreover, we believe that this correction framework can be extended to any classification task, beyond object detection.

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

# A APPENDIX

## A.1 PSEUDO-CODE FOR AUTOENCODER TRAINING

We present the pseudo-code for autoencoder training that includes the weight reparameterization trick in Alg. 4.

---

**Algorithm 1** Autoencoder training

---

**Input:** Number of Gaussians $K$, Latent Samples $\left\{z_i^{\text{FP}}\right\}_{i=1}^{N}$ and Bounding Box Samples $\left\{y_i^{\text{box, FP}}\right\}_{i=1}^{N}$ of FP in $\mathcal{D}_{\text{C}}$, the number of samples to sample per Gaussian $m$.
**Output:** GMM parameters $\{\pi_i, \mu_i, \Sigma_i\}_{i=1}^{K}$ optimized, and the parameters $\Theta$ of the Decoder $\Phi_\Theta$ optimized.

1: $\mathcal{L}_{\text{GMM}} \leftarrow \infty$
2: Initialize $\{\pi_i, \mu_i, \Sigma_i\}_{i=1}^{K}$ to optimize a GMM over $\left\{z_i^{\text{FP}}\right\}_{i=1}^{N}$ thanks to EM algorithm
3: **while** $\mathcal{L}_{\text{Autoencoder}}$ has not converged **do**
4: $\quad$ $\boldsymbol{z} \leftarrow$ Empty Tensor
5: $\quad$ $\boldsymbol{y}_{\text{bbox}} \leftarrow$ Empty Tensor
6: $\quad$ $\boldsymbol{y}_{\text{class}} \leftarrow$ Empty Tensor
7: $\quad$ **for** i=1 to K **do**
8: $\quad\quad$ Sample $m$ times from $\mathcal{N}(\mu_i, \Sigma_i)$ with the reparameterization trick: $\left\{z_j^{(i)}\right\}_{j=1,..,m}$ and concatenate with $\boldsymbol{z}$
9: $\quad\quad$ Compute $\left\{y_{j,bbox}^{(i)} = \sigma \circ \text{MLP}_{\text{bbox}}\left(z_j^{(i)}\right)\right\}_{j=1,..,m}$ and concatenate with $\boldsymbol{y}_{\text{bbox}}$
10: $\quad\quad$ Compute $\left\{y_{j,class}^{(i)} = \text{MLP}_{\text{class}}\left(z_j^{(i)}\right)\right\}_{j=1,..,m}$ and concatenate with $\boldsymbol{y}_{\text{class}}$
11: $\quad$ **end for**
12: $\quad$ Sample $m \times K$ times from $\left\{y_i^{\text{bbox, FP}}\right\}_{i=1}^{N}$: $\left\{y_{\tau(i)}^{\text{bbox, FP}}\right\}_{i=1}^{m \times K}$
13: $\quad$ Compute $\mathcal{L}_{\text{GMM}} = \mathcal{W}\left(\sum_{i=1}^{K} \frac{\pi_i}{m} \sum_{j=1}^{m} \delta_{y_{j,bbox}^{(i)}}, \frac{1}{m \times K} \sum_{i=1}^{m \times K} \delta_{y_{\tau(i)}^{\text{bbox, FP}}}\right) + \mathcal{L}_{\text{CE}}\left(\varnothing, \boldsymbol{y}_{\text{class}}\right)$
14: $\quad$ Compute $\mathcal{L}_{\text{Decoder}} = \mathcal{L}_{\text{MSE}}\left(\Phi_\Theta(\boldsymbol{y}_{\text{bbox}}), \boldsymbol{z}\right)$
15: $\quad$ Compute $\mathcal{L}_{\text{AutoEncoder}} = \mathcal{L}_{\text{GMM}} + \mathcal{L}_{\text{Decoder}}$
16: $\quad$ Gradient step over $\mathcal{L}_{\text{AutoEncoder}}$ with respect to $\{\pi_i, \mu_i, \Sigma_i\}_{i=1}^{K}$ and $\Theta$
17: **end while**

---

Figure 4: Pseudo-code for autoencoder training.

## A.2 PROOF OF THE WEIGHT REPARAMETERIZATION TRICK

The weight reparameterization trick is based on the law of total probability. Indeed, let $\boldsymbol{Z} \sim \text{GMM}\left(\{\pi_i, \mu_i, \Sigma_i\}_{i=1}^K\right)$ and $\boldsymbol{Y} = f(\boldsymbol{Z})$ where $f$ should be $\sigma \circ \text{MLP}_{\text{bbox}}$ or $\text{MLP}_{\text{class}}$. We have

$$P\left(\boldsymbol{Y} = y\right) = P\left(f(\boldsymbol{Z}) = y\right) \tag{11}$$

$$= \sum_{i=1}^K P\left(f(\boldsymbol{Z}) = y | \boldsymbol{Z} \sim \mathcal{N}(\mu_i, \Sigma_i)\right) P(\boldsymbol{Z} \sim \mathcal{N}(\mu_i, \Sigma_i)) \tag{12}$$

$$= \sum_{i=1}^K \pi_i P(f(\boldsymbol{Z}) = y | \boldsymbol{Z} \sim \mathcal{N}(\mu_i, \Sigma_i)) \tag{13}$$

Therefore, if we sample $m$ times from each $K$ Gaussians $\{z_j^{(i)}\}_{1 \leq j \leq m}^{1 \leq i \leq K}$, the discrete distribution $\alpha$ in the final space (of logits or bounding boxes) is:

$$\alpha = \sum_{i=1}^K \sum_{j=1}^m \frac{\pi_i}{m} \delta_{f\left(z_j^{(i)}\right)} \tag{14}$$

Since $\{\pi_i\}_{1 \leq i \leq K}$ appears in the discrete distribution and consequently in the expression of the Wasserstein loss, we can now backpropagate this loss over the weights $\{\pi_i\}_{1 \leq i \leq K}$.

## A.3 DESIGN CHOICES FOR THE AUTOENCODER

We chose to train the GMM and the decoder simultaneously using an autoencoder, though this approach might seem suboptimal at first. Indeed, the GMM initially fails to generate suitable training samples for the decoder, leading the latter to learn from inaccurate samples during the first epochs. A potential alternative is to first train the GMM separately, followed by training the decoder with the GMM frozen. This method is expected to improve stability and accelerate the overall process. Nevertheless, as demonstrated in Fig. 5, this approach does not work as intended. While the GMM performs well and generates target TNs for similar FPs in $\mathcal{D}_\text{C}$, the decoder struggles to retrieve the embedding. The inability to correctly predict the position of the embedding of a TN given a bounding box is due to the average variance of Normal distributions that compose the GMM (which can be defined as $\frac{1}{K} \sum_{i=1}^K \left(\sum_{j=1}^{\dim(\mathcal{Z})} [\Sigma_i]_{j,j}\right)$) which produces suitable samples but spans a large region in $\mathcal{Z}$. Consequently, the decoder faces an overly difficult task, as two samples generated by the GMM could be far apart in $\mathcal{Z}$ despite having nearly identical bounding boxes. Interestingly, when the GMM is initialized with the current latent distribution of FPs in $\mathcal{D}_\text{C}$, it shows low variance, indicating the feasibility of achieving a more focused distribution.

Thus, training the GMM and decoder together within an end-to-end autoencoder introduces a form of regularization on the GMM's variance. As shown in Fig. 6 the GMM is initially optimized with high variance, followed by the decoder's optimization, which adapts to the GMM's decreasing variance. This process can be interpreted as: after identifying large regions containing target TNs, the GMM then narrows down its focus within those regions.

## A.4 IMPLEMENTATION DETAILS

**DETR** We present the hyperparameters that have been employed during the training and the correction of DETR model in 6. Unless otherwise noted, the hyperparameters remain consistent during both training and correction phases. During the guide step, the model is exclusively fed with images that contain instances of the confused class. The training set is split into two subsets, reserving 30% of the data as a validation set.

**Autoencoder** To parameterize the GMM, certain established techniques must be employed. First, we optimize the unnormalized weights, ensuring the normalization condition $\sum_{i=1}^K \pi_i = 1$ by applying the Softmax function. A covariance matrix must be a symmetric positive definite matrix, and

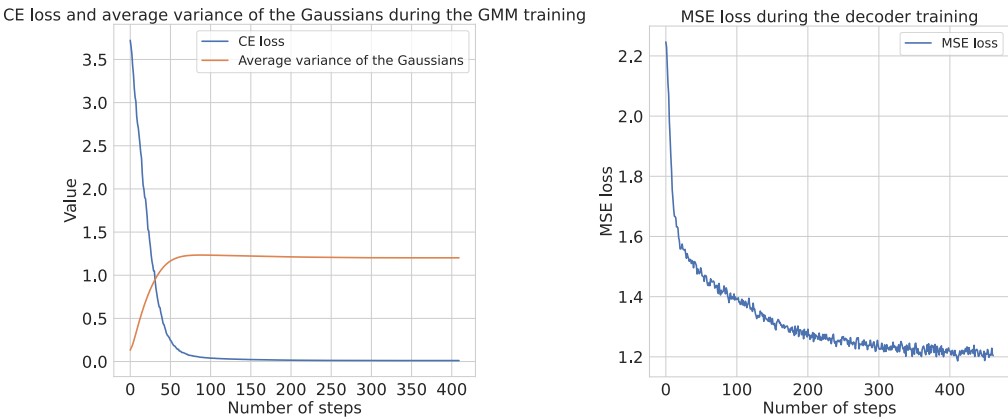

Figure 5: Evolution of the losses and the average variance of the Gaussians when we train separately the GMM and then the decoder.

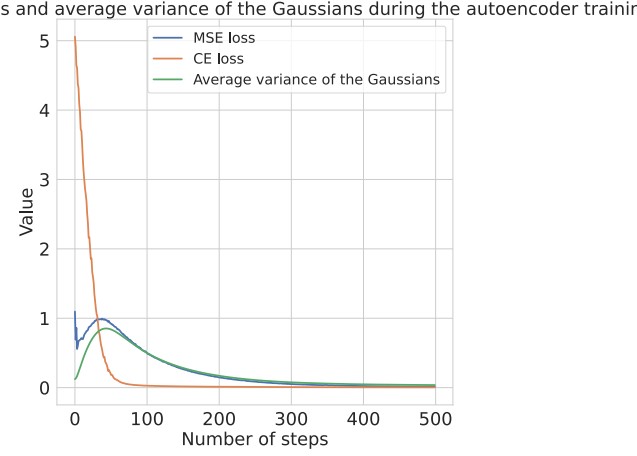

Figure 6: Evolution of the losses and the average variance of the Gaussians when we train the GMM and the decoder simultaneously.

since all symmetric positive matrices possess a square root, we train the square root matrix and subsequently derive the covariance matrix by multiplying it by its transpose. To maintain the positive definiteness, we add $1.10^{-4}$ to all diagonal elements. The architecture of the decoder is detailed in Tab. 7

Table 6: Hyperparameters during the training and the correction.

| Hyperparameter | Value |
|---|---|
| Batch size | 4 |
| Gradient accumulation | 8 |
| Early stopping criterion (training) | If the validation loss does not decrease over 10 epochs |
| Optimizer (training) | AdamW; $\text{lr}_{\text{backbone}} = 1e-5$; $\text{lr}_{\text{rest}} = 1e-4$; weight_decay $= 1e-4$ |
| Weight decay | $1e-4$ |
| Learning step scheduler | Divide by 2 if the validation loss does not decrease over 5 epochs |
| Guidance stopping criterion | If the validation correction loss does not decrease over 5 epochs |
| Repair stopping criterion | If the validation loss does not decrease over 5 epochs |
| Optimizer (correction) | AdamW; $\text{lr}_{\text{backbone}} = 1e-6$; $\text{lr}_{\text{rest}} = 1e-5$; weight_decay $= 1e-4$ |
| Temperature $T$ in $\mathcal{L}_{\text{Guide}}$ in LaGF | $T = 1$ |

Table 7: Architecture of the decoder.

**Decoder**

Input $\in \mathbb{R}^4$

Linear(4, 256)

Linear(256, 256)

Linear(256, 256)

Output $\in \mathbb{R}^{256}$

