# OpenReview forum: "ReFOCUS: Recurrent False Object Correction Using guidance Strategies in Object Detection"
_ICLR.cc/2025/Conference — Submitted to ICLR 2025_

### Official Review · Reviewer_yPBg · 2024-10-31

**Soundness:** 3
**Presentation:** 2
**Contribution:** 2
**Rating:** 5
**Confidence:** 3

**Summary:**

This paper aims to correct false positives in object detection. Starting from erroneous annotations and non-bojects similar to the true samples, the model is corrected in the latent and logit spaces.

**Strengths:**

1. The paper provides a rich theoretical basis and source of motivation.
2. The paper verifies the effectiveness of the proposed method in both general object detection and few-shot object detection scenarios.

**Weaknesses:**

1. The paper mentions corrections in two spaces in both the abstract and contributions, but lacks a description of logit.
2. The paper mentions that ‘we assume that we have access to a corrective dataset DC’. It seems that the paper's method relies on clear FP data with the same data domain as the initial dataset to guide GMM generation and model fine-tuning, but such data is often not easy to obtain. We often cannot access the data after training a network, not to mention the FP information. How to solve this?
3. The paper proposes multiple loss functions in the method stage. It is recommended that the paper mark the different stages and training parameters of each loss in the Figure to facilitate readers' understanding. For example, in the final loss L_decoder stage, the author's network contains two MLP_bbox modules. Which one does ybbox target?
4. In Table 2, it seems that the author's method does not improve significantly compared to directly using f_Ture. Did the author compare the extra cost of this method compared to f_True? For example, a time comparison for such a two-stage process.

**Questions:**

Before the Definition 1, the paper set the TN ‘only its position remains to be defined’, but after Definition 1, the paper said the TN ‘shares the same bounding box as the FP.’ How does this information align?

---

> ### Author Response · Authors · 2024-11-16
>
> Thank you for your extensive reviews and the valuable feedback. We have addressed each of your comments individually below. Before responding to specific questions and concerns, we would like to clarify the motivations and context of our work.
>
> After training an object detection model, users may notice certain recurring false positive (FP) errors, where similar background objects are wrongfully detected. In our work, we don't commit on a specific appearance threshold for errors to be considered as recurring. We rather focus on developing an adaptable framework that can be applied to most specific scenarios, whether the user frequently notices similar FP post-training, or only sporadically.
>
> This study considers two scenarios leading to such errors: (1) the dataset may be poorly annotated, causing the model to misidentify background as FP objects due to training biases, or (2) the dataset may be well-annotated, but FP objects visually resemble actual target objects, prompting detection errors. The second scenario often resembles an overfitting issue specific to a use case. For instance, in pedestrian detection, the model might detect people inside cars as pedestrians. This work aims to leverage annotated images of people in cars to train the model to distinguish pedestrians from passengers in vehicles.
>
> For several reasons outlined in the introduction of the paper, the methods need to support continuous fine-tuning.
>
> To enhance clarity on our motivations, we will further elaborate on this in the revised version.
>
> **Q1)** *Lack of a description of logit.*
>
> We elaborate more on LaGF than on LoGF because 1) LaGF has better results and 2) LaGF features new concepts. While LaGF involves estimating a target embedding for each FP, LoGF is a simpler approach that only up-weights specific background objects in the cross-entropy loss.
>
> **Q2)** *Difficulty in getting the corrective dataset D_C.*
>
> This work explores utilizing FP annotations specifically available after training, so while access to clear in-domain FP data is necessary here, access to the entire training dataset $\mathcal{D}_\text{True}$​ is not, as highlighted in our few-shot analysis. Future work will explore 1) domain-shifted FP data 2) methods that do not require additional annotations. Our end goal is to use LaGF and LoGF to solve recurring misclassifications by shifting objects from a commonly confused class to their correct class, and not only to background.
>
> **Q3)** *In the final loss L_decoder stage, the author's network contains two MLP_bbox modules. Which one does ybbox target?*
>
> We are working to clarify this in the revised version. As noted in line 334, $y_\text{bbox} = \sigma \circ \text{MLP}_\text{bbox}(z)$, where $z$ is the embedding generated by the GMM, and $\sigma$ represents the sigmoid function. There is only one MLP_bbox module which is shared across all embeddings, but the L_decoder is only computed and backpropagated over FP embeddings.
>
> **Q4)** *In Table 2, it seems that the author's method does not improve significantly compared to directly using f_Ture. Did the author compare the extra cost of this method compared to f_True? For example, a time comparison for such a two-stage process.*
>
> In Table 2, both LaGF and LoGF significantly improve the noisy model $f_\text{Noisy}$, even achieving better results than $f_\text{True}$​. However, we indeed observe limited improvement when applied/compared to the already accurate model $f_\text{True}$, which makes sense, as $f_\text{True}$ already performs well and has fewer errors to correct.
> Under the assumption of a noisy initial dataset, using LoGF and LaGF instead of $f_\text{True}$​ offers multiple advantages, since these methods support continuous fine-tuning without needing the entire dataset to be re-annotated, unlike $f_\text{True}$​. Then, LoGF and LaGF can be applied to a dataset subset without erasing knowledge from the initial training, which brings environmental and time benefits by reducing the number of operations needed for convergence. Thus, the main advantage of using LoGF and LaGF over $f_\text{True}$ is to leverage the benefits of continuous fine-tuning, as outlined in the introduction. Another advantage is enabling post-training correction to fine-tune for a specific use case, as noted in the motivations at the beginning of this rebuttal.
>
> Time comparisons are definitely missing, but they are challenging, as they depend on factors like the number of classes, the frequency of the confused classes in the dataset, and the chosen subset size. However, training the autoencoder is efficient due to the limited number of parameters, and the guide and repair steps resemble typical fine-tuning in terms of time requirements and operations

---

> > ### Comment · Reviewer_yPBg · 2024-11-28
> >
> > Thank you for your time and response. I understand the authors’ response and efforts, but I didn't find corresponding response to my Questions.

---

### Official Review · Reviewer_MRKq · 2024-11-02

**Soundness:** 2
**Presentation:** 2
**Contribution:** 2
**Rating:** 3
**Confidence:** 3

**Summary:**

This work addresses the issue of recurrent false positive (FP) classification in object detection. The paper proposes two innovative correction frameworks that guide FPs toward TNs in either the latent space or the logit space. The latent guidance framework leverages an autoencoder where a learnable Gaussian mixture model generates the embeddings of appropriate TNs, and a straightforward decoder retrieves the TN embedding given a bounding box. The paper utilizes the properties of the Wasserstein distance to train the Gaussian mixture model through standard backpropagation.

**Strengths:**

1. The experiments show that the propsoed module improves the performance.
2. The idea of introducing other spaces other than logit space to solve classification errors is valuable.

**Weaknesses:**

1. Figure 1 is the network diagram in DETR paper, without any information increment, so this content should not appear in the main text. In addition, although illustrations of motivation may not be necessary in the Introduction section, the description of the motivation section in this article should have an image which includes:
(a) the case visualization and explanation of the problem of false positives in classification errors.
(b) the comparison with previous practices, as said in 'Motivations' part in Line 68. Tell the reader how the proposed method addresses the problem compared to previous approaches.
(c) the visualization of the noisy dataset D_{Noisy}, and well-annotated dataset D_{True}. And how they are used in the method.

2. Line 062 said the method 'can be generalized across different datasets and detection frameworks.' However, there were no experiments conducted on additional datasets and detectors in the experimental section and supplementary materials. In addition, both the data volume and the number of categories of VOC07 are not representative enough. Can it be validated with a larger dataset  (e.g., COCO, OpenImages)  and extra detectors to  better demonstrate the method's generalizability? Also, could you provide the computational requirements for testing on larger datasets if that is limited in larger dataset experments.

3.  The writing and paper organized (like the structure of method section) should be improved. For example:
(a). Sec 3.3.1 has 'Definition 1' (Line 221), then what? There are no  'Definition 2' in the following part.
(b). The relationship between sections in method. You can provide  an overview of  'LoGF' and 'LaGF' in 'CORE CONCEPT' part and tell us the structure of the following sub-sections.
......
and, there are too many colloquial words in the article, such as "we" appearing 107 times.

4. More visual displays are necessary. You don not need to show the results here, but rather make the before-and-after comparisons of specific false positive cases, or visualizations of how the latent space changes with the proposed methods.  In addition, some qualitative experiments on false positives can be added, such as analyzing whether the proportion of false positives decreases after adding methods from multiple dimensions, and so on.

**Questions:**

See weakness part.

---

> ### Author Response · Authors · 2024-11-16
>
> Thank you for your review and comments. We have addressed each of your comments individually below.
>
> **Q1)** *DETR Figure removal and additional visualization.*
>
> Figure 1 will be removed in the revised version. The network was initially included to improve understanding, but it will be replaced with illustrations of the different datasets and the different outputs, showing both the expected results and actual outcomes in selected cases. The previous approaches—standard fine-tuning and retraining from scratch—are also compared in the experiments to highlight the benefits of our method. The two datasets, $\mathcal{D}_\text{Noisy}$​ and $\mathcal{D}_\text{True}$​, are used to train the base model, resulting in $f_\text{Noisy}$ ​and $f_\text{True}$​, which are subsequently enhanced by LoGF and LaGF. The corrective dataset $\mathcal{D}_\text{C}$ will be also presented, as it is the actual dataset used in the methods. In addition, to clarify the motivation and context of this work, expected outputs and actual outputs will be added.
>
> **Q2)** *Method's generalizability?*
>
> This paper introduces a new concept—post-training correction of a model—which involves a substantial amount of foundational context and methodology. Conducting additional experiments on other datasets and detectors could not be carried out in time and within the main paper due to the limited number of pages. Some additional experiments could be included in the appendix.
>
> In this paper, we conduct experiments on two subsets of VOC2007 (sofa/chair and car/bus), which are chosen based on category hierarchy as you can see in Fig.2 in [1] and form two non-overlapping single-class datasets. We believe this design provides evidence against overfitting, demonstrating the method's capability without a more complex dataset. Future work will aim to validate the method further using different detectors as well as larger and more diverse datasets such as COCO or OpenImages.
>
> Time comparisons are challenging, as they depend on factors like the number of classes, the frequency of the confused classes in the dataset, and the chosen subset size. However, training the autoencoder is efficient due to the limited number of parameters, and the guide and repair steps resemble typical fine-tuning in terms of time requirements and operations. Since LoGF and LaGF allow continuous fine-tuning, they enable the use of a subset of a large dataset, while the efficiency of the corrections mainly depends on the number of images containing FPs.
>
> **Q3)** *The writing and paper organized (like the structure of method section) should be improved.*
>
> We have taken this comment into account and we are making improvements in the revised version. Specifically, the use of "Definition 1" has been removed. We are also adding a clearer outline in the "CORE CONCEPT" subsection to better describe the structure of the following sections. While the "CORE CONCEPT" paragraph already provides an overview of LoGF and LaGF, we will enhance this part in the revised version for better clarity.
>
> **Q4)** *More visual displays are necessary. You do not need to show the results here, but rather make the before-and-after comparisons of specific false positive cases, or visualizations of how the latent space changes with the proposed methods. In addition, some qualitative experiments on false positives can be added, such as analyzing whether the proportion of false positives decreases after adding methods from multiple dimensions, and so on.*
>
> Visualizing the changes in the latent space is a valuable direction for future work. This process requires significant efforts in clustering and understanding how the latent space evolves, which involves a more in-depth investigation. The DETR framework sets aside N (typically N=100) object queries, which specialize in detecting specific classes of objects in specific positions. This further emphasizes the need for larger datasets to explore the latent space thoroughly.
>
> In our current work, we focus on mAP as a more stable metric than the false positive proportion, though the latter is indeed an interesting measure. For more granular analysis, we examine Precision and Recall in the single-class case to track the evolution of false positives (FP), true positives (TP), and true negatives (TN), which are the most relevant metrics for this context, as discussed in Lines 380-381.
>
> The current version is actually being revised, taking into account your feedback and that of the other reviewers. You are helping to improve the paper and we thank you for that. We will submit the revised version as soon as possible.
>
>
> [1] M. Everingham, L. Van Gool, C. K. I. Williams, J. Winn, and A. Zisserman. The
> PASCAL Visual Object Classes Challenge 2007 (VOC2007) Results. http://www.pascal-
> network.org/challenges/VOC/voc2007/workshop/index.html.

---

> > ### Comment · Reviewer_MRKq · 2024-11-25
> >
> > I I still believe that validation on larger datasets is necessary, which makes the contribution proposed in the article more convincing for publication at the top conference. Additionally, is there any formatting error in Q1's response?

---

### Official Review · Reviewer_MM1g · 2024-11-03

**Soundness:** 2
**Presentation:** 2
**Contribution:** 3
**Rating:** 5
**Confidence:** 4

**Summary:**

The work focuses on guiding the false positive object predictions to true negative predictions. The paper shows that models with FP predictions can be efficiently corrected using FP annotations. The paper proposed two correction approaches that guide false positives toward true negatives, i.e., in latent space (LoGF) and in logit space (LaGF). Both two methods required a corrective dataset where all recurrent FPs are additionally annotated. LoGF only modifies the classification logits of FP. LaGF uses an autoencoder architecture to change FP latents (sampled in a trained GMM) to TN latents (decoded by a trained decoder). After the training of the decoder, the original detection model is guided with newly annotated FP labels and  the decoder to bridge the gap between FP and TN in latent space, therefore, avoiding the FP predictions.

**Strengths:**

The paper explores a new area to show that models with FP predictions can be efficiently corrected using FP annotations with two proposed methods, LoGF and LaGF.

**Weaknesses:**

1. Unfair comparison. While these two proposed guidance frameworks both rely on a corrective dataset where all recurrent FPs are additionally annotated, a fair comparison should be a noisy model fine-tuned on the combination of corrective dataset D_c and correct dataset D_{True} instead of fined-tuned only on correct dataset D_{True}.
2. The experiments only explore the situation when only one pair of classes are misannotated, however, in the real world, multiple misannotated pairs are more common.
3. Some mistakes in writing. In Section 4.1, the authors use PASCAL VOC 2007, while in Section 5.1, the authors say to train the model on MS-COCO 2017. And no labels on the caption of tables to point out whether the results are done on the former or the latter dataset.

**Questions:**

1. Please refer to the weakness.
2. Why contrastive learning is listed as one subsection of related works?

---

> ### Author Response · Authors · 2024-11-16
>
> Thank you for your review and valuable comments. Some changes are currently in progress and we will submit the revised version as soon as possible. We have addressed each of your comments individually below.
>
> **Q1)** *Unfair comparison. While these two proposed guidance frameworks both rely on a corrective dataset where all recurrent FPs are additionally annotated, a fair comparison should be a noisy model fine-tuned on the combination of corrective dataset D_c and correct dataset D_{True} instead of fined-tuned only on correct dataset D_{True}.*
>
> We did perform a comparison by fine-tuning the model on $\mathcal{D}_\text{C}$, which involved adding a new class to store the false positives, as we also did with the oracle model. However, this approach led to significant performance degradation and very high variance. Resetting the classification head (by adding an additional class) completely disrupted the model, necessitating a complete retraining. Even when fine-tuning on the entire dataset, the results still showed considerable degradation. Given the poor quality of these results, we decided not to include them in the final study.
>
> **Q2)** *The experiments only explore the situation when only one pair of classes are misannotated, however, in the real world, multiple misannotated pairs are more common.*
>
> This paper serves as a foundational exploration of the concept of classification correction after model training by leveraging FP annotations. It is a comprehensive work, but not exhaustive, focusing on establishing a new context as well as methods for utilizing FP annotations. While the current experiments concentrate on cases with a single misannotated pair of classes, we did attempt to misannotate relevant classes that could lead to such errors. Specifically, we selected pairs of classes from the same category in the VOC2007 hierarchy, as outlined in Fig. 2 in [1].
>
> **Q3)** *Some mistakes in writing. In Section 4.1, the authors use PASCAL VOC 2007, while in Section 5.1, the authors say to train the model on MS-COCO 2017. And no labels on the caption of tables to point out whether the results are done on the former or the latter dataset.*
>
> We use the DETR model, pretrained on MS-COCO 2017, as the base model. Afterward, we fine-tune this model on $D_\text{Noisy}$ to obtain $f_\text{Noisy}$​, on $D_\text{True}$ to obtain $f_\text{True}$, and so on. All experiments, however, are conducted on PASCAL VOC 2007. The decision to use DETR pretrained on MS-COCO 2017 stems from several reasons:
>
> Fine-tuning a pretrained model is less time-consuming.
> It yields better performance on PASCAL VOC 2007, ensuring that our frameworks are tested under optimal conditions, where the model is already strong or at least difficult to improve further through retraining.
> Pretrained weights help mitigate high variance during training, which is crucial for the object detection task.
> We will clarify these points in the revised version to prevent any confusion.
>
> **Q4)** *Why contrastive learning is listed as one subsection of related works?*
>
> Contrastive learning was explored extensively during the study, as the objective was to bring false positives (FPs) closer to true negatives (TNs) in the latent space. Initially, we also considered the possibility of pushing FPs away from TPs. However, in the final version of the work, we found that contrastive learning was not as relevant to our approach as we had originally thought. As a result, we will remove this subsection in the revised version to better align with the final direction of our work.
>
>
>
>
> [1] M. Everingham, L. Van Gool, C. K. I. Williams, J. Winn, and A. Zisserman. The
> PASCAL Visual Object Classes Challenge 2007 (VOC2007) Results. http://www.pascal-
> network.org/challenges/VOC/voc2007/workshop/index.html.

---

### Official Review · Reviewer_7k3H · 2024-11-08

**Soundness:** 2
**Presentation:** 1
**Contribution:** 2
**Rating:** 3
**Confidence:** 5

**Summary:**

In this paper, two strategies of improving object detectors are proposed to handle the issue caused by erroneous annotations and false positives. It is achieved by guiding the false positives toward true negatives in the latent or the logit space. By further fine-tuning on correct annotations, the proposed methods improve the detection performance in most cases.

**Strengths:**

1. Two strategies of improving object detectors are proposed.
2. Training details are provided to ensure its reproducibility.

**Weaknesses:**

1. The writing is poor. The authors need to carefully revised the paper from aspects of logical structures and grammars.
2. In the first sentence of abstract, “recurrent false positive classification” is required to be further explained. In the first paragraph of introduction, the explanation of “recurrent errors” is also needed. Are these two concepts have the same meaning?
3. The motivation may be questionable. In introduction, the authors claim that “the model consistently detects an object that should not be identified, e.g. people on billboards as instances of real people.” However, the phenomenon that people on billboards are detected may be reasonable in some situations. Classifying these samples to be background by force could increase the risk of model oscillation or overfitting.
4. In introduction, the Motivation paragraph seems to be logically incorrect. It seems that D_{True} is used in all cases. Why not using the f_{True} directly? A detailed explanation of this paragraph is required. Furthermore, the proposed methods only slightly outperform f_{True}. In some cases, the performance is even dropped, which limits the contribution of this paper.
5. The experiments are not convincing. For PASCAL VOC dataset, there are 20 classes of objects. However, the authors only uses a small part of them. The mAP on all classes should be reported.

**Questions:**

The motivation of this paper should be re-clarified. The complete experimental results should be reported. The overall contribution is limited. Furthermore, this paper needs a major revision by re-organizing and proofreading its paragraphs.

---

> ### Author Response · Authors · 2024-11-16
>
> Thank you for your review and comments. The whole grammar and structure are currently being revised and we will submit the revised version as soon as possible. We have addressed each of your comments individually below.
>
> **Q1)** *Definition of 'recurrent false positive classification"*
>
> The definitions of recurrent errors, and specifically recurrent false positive classification, are clarified in the revised version. To make it clear, a recurrent error refers to the same type of error occurring multiple times, or often enough that it becomes noticeable. Interestingly, there is no fixed threshold for frequency; LoGF and LaGF can be applied even when the model works well overall. For example, in the car/bus dataset, where only cars are annotated and buses are not (referencing D_true in the paper), the model may classify buses incorrectly as cars in a few instances. When applying LaGF to this model with cars and buses, the incorrectly classified bus (FP) would be shifted towards a T. The other buses, already labeled as TNs, would either stay as they are or be moved to a higher-confidence TN (lines 260-263). Although applying LaGF may slightly disturb the performance of other classes, the experiments show that the impact is minimal. If significant degradation occurs, a future direction could involve moving FPs towards soft TNs—using, for example, confidence thresholds or adjusting the CE loss during autoencoder training. This interesting approach will be explored in future works.
>
> Recurrent false positive classification is a specific type of recurrent error, where objects that should belong to the background are mistakenly detected as foreground objects.
>
> **Q2)** *The motivation may be questionable.*
>
> The motivation will be clarified in the revised version. The correction of such errors arises from observations made by external users. For instance, if the use case of the detection model is to identify pedestrians on the street, and users notice that people inside vehicles or on billboards are also being detected, they may wish to exclude these detections. In such cases, after annotating images with people inside vehicles or on billboards that should not be detected, applying LaGF or LoGF can be highly beneficial. By annotating these specific examples, there is hope that the model will learn the context (e.g., the difference between pedestrians and people inside vehicles or on billboards) and generalize accordingly, reducing the risk of oscillation or overfitting. The primary goal of this method is to be an industrial solution applied after user observations, rather than just a way to improve the state-of-the-art on public datasets without real-world objectives.
>
> **Q3)** *Why not using the f_{True} directly?*
>
> The motivation paragraph will be clarified to address any misunderstandings. As mentioned in the motivation section, directly training on $f_\text{True}$ might not always be feasible (data might no longer be available, excessive computational cost...). However, LoGF and LaGF do not require the complete dataset D_True with FP annotations​, but only a subset of it. This approach allows for continuous fine-tuning of the model on only a part of the dataset, preserving the knowledge from the initial training while correcting the errors.
>
> Thus, using LoGF and LaGF instead of directly using $f_\text{True}$​ offers several advantages: it requires only a subset of the dataset, making the training less time-consuming and reducing human effort. Additionally, in cases where the model has been trained on a correctly annotated dataset but still makes errors based on the use-case, LoGF and LaGF offer the flexibility to perform customized corrections based on user observations.
>
> The improvements on $f_\text{True}$​ in the bus/car dataset are modest because the base model already performs well in this case, correctly classifying the buses as background objects. However, we demonstrate the robustness of LaGF, which consistently improves the model by correcting the rare instances where buses are incorrectly classified as cars.
>
> **Q4)** *The experiments are not convincing.*
>
> This paper is intended as a foundational work and is not exhaustive. Testing on all 20 classes of the PASCAL VOC dataset would require significant additional experimentation, which we plan to explore in future work. For this foundational study, we focused on two relevant pairs of objects, chosen because they belong to the same category in the hierarchy of the PASCAL VOC 2007 dataset, as outlined in Fig. 2 in [1].
>
> In response to the comment, we have added the mAP results for all classes in the revised version, which will provide a more comprehensive evaluation of the proposed method.
>
> [1] M. Everingham, L. Van Gool, C. K. I. Williams, J. Winn, and A. Zisserman. The PASCAL Visual Object Classes Challenge 2007 (VOC2007) Results. http://www.pascal-network.org/challenges/VOC/voc2007/workshop/index.html.

---

> > ### Comment · Reviewer_7k3H · 2024-11-28
> >
> > Thanks for your response to my review comments. My concerns regarding Q1 and Q3 have addressed. However, for Q2, I still find the motivation of the paper insufficiently strong, which limits the applicability and contribution of the proposed method. As for Q4, while the authors have committed to including more comprehensive experiments in the revised version, no supporting data has been provided. Combined with the lack of sufficient and necessary experimental analysis in the current paper, I have decided to maintain my original score.

---

### Meta-Review · Area_Chair_wqHb · 2024-12-19

**Metareview:**

The goal of the paper is to improve object detectors using two approaches: LoGF (latent space) and LaGF (logit space). These methods guide false positives  toward true negatives by modifying classification logits or using an autoencoder to change false positives  latents to rue negatives latents. By fine-tuning on correct annotations, the proposed methods can efficiently correct models with false positives  predictions. Reviewers seem to agree that the latent space method is novel and like the theoretical aspects of the paper. However, all reviewers seem to have comments regarding the writing and presentation of the paper. Most are also not convinced about the experiments. Hence the paper is not ready in its current form for ICLR.

**Additional Comments On Reviewer Discussion:**

The rebuttal did not move the scores for any of the reviewers. Reviewer 7k3H was promised more experiments but no data was included. Reviewer MM1g was not satisfied with the use of other datasets.  Reviewer MRKq thought that the datasets used were and  Reviewer yPBg did not get responses to their question.

---

### Decision · Program_Chairs · 2025-01-22

Reject